# MoE-PHDS: One MoE checkpoint for flexible runtime sparsity

## Abstract

Sparse Mixtures of Experts (MoEs) are typically trained to operate at a fixed sparsity level, e.g. $k$ in a top-$k$ gating function. This global sparsity level determines an operating point on the accuracy/latency curve; currently, meeting multiple efficiency targets means training and maintaining multiple models. This practice complicates serving, increases training and maintenance costs, and limits flexibility in meeting diverse latency, efficiency, and energy requirements. We show that pretrained MoEs are more robust to runtime sparsity shifts than commonly assumed, and introduce MoE-PHDS (**P**ost **H**oc **D**eclared **S**parsity), a lightweight SFT method that turns a single checkpoint into a global sparsity control surface. PHDS mixes training across sparsity levels and anchors with a short curriculum at high sparsity, requiring no architectural changes. The result is predictable accuracy/latency tradeoffs from one model: practitioners can "dial $k$" at inference time without swapping checkpoints, changing architecture, or relying on token-level heuristics. Experiments on OLMoE-1B-7B-0125, Qwen1.5-MoE-A2.7B, and proprietary models fit on multiple operating points show that PHDS matches or exceeds well-specified oracle models, improves cross-sparsity agreement by up to 22% vs. well-specified oracle models, and enables simplified, flexible runtime MoE deployment by making global sparsity a first-class serving primitive.

## 1 Introduction

Mixture of Experts (MoEs) language models deliver state-of-the-art quality with lower active compute by routing tokens through a subset of active experts per layer (Liu et al., 2024; Du et al., 2022). At deployment, sparsity level (e.g. $k$ in top-$k$) is fixed, so supporting multiple operating points has required multiple checkpoints. We argue this is not necessary. First, pretrained MoEs already tolerate moderate runtime sparsity shifts. Second, with MoE-PHDS we make this tolerance more predictable: a short SFT schedule across sparsity levels with curriculum anchoring produces a single checkpoint reusable at different sparsity levels. This enables more predictable accuracy/latency tradeoffs, SLA-aware serving, and energy-aware throttling from one model.

Prior work, like adaptive routing(Huang et al., 2024; Nishu et al., 2025; Alizadeh-Vahid et al., 2024) and null experts (Zeng et al., 2024; Yan et al., 2025; Team et al., 2025), focuses on token-level choices. These can spend a fixed global budget more efficiently, but add variance (latency depends on input) and policy complexity (extra knobs). In contrast, PHDS exposes *one global knob: $k$*, which yields more predictable accuracy/latency tradeoffs. This simplicity is key for deployment.

Prior work in adaptive computation, such as slimmable networks (Yu et al., 2019), Once-for-All models (Cai et al., 2020), MatFormer (Devvrit et al., 2024), and Flextron (Cai et al., 2024), demonstrates that dense networks can support multiple operating points from a single model. In contrast, the prevailing approach is that each global MoE sparsity level needs its own checkpoint, often from a separate pretraining run. This practice of supporting only well-specified models leads to training and storing multiple models, and has stymied study about model behavior when inference sparsity is not well-specified. As such, fundamental questions remain unanswered: *Can a single MoE checkpoint generalize across multiple global levels of sparsity? What mechanisms block or enhance model flexibility?*

Our work challenges the fixed-sparsity assumption. We show that MoEs are robust to small-to-moderate changes in the global sparsity parameter. Empirically, we show that flexibility can be

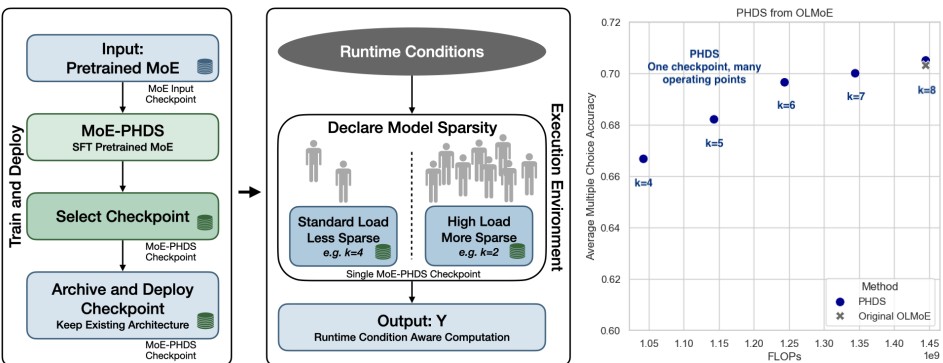

Figure 1: Overview of our proposed framework, MoE-PHDS. The left represents how a robust, sparsity-flexible single checkpoint is generated. The center shows how the model is called under dynamic runtime conditions. The right panel shows average multiple choice task accuracy vs. flops with 4096 context length using OLMoE-1B-7B-0125 as a base pretrain model, along with the a well-specified model. There is little accuracy degradation as $k$ is reduced from 8 to 6.

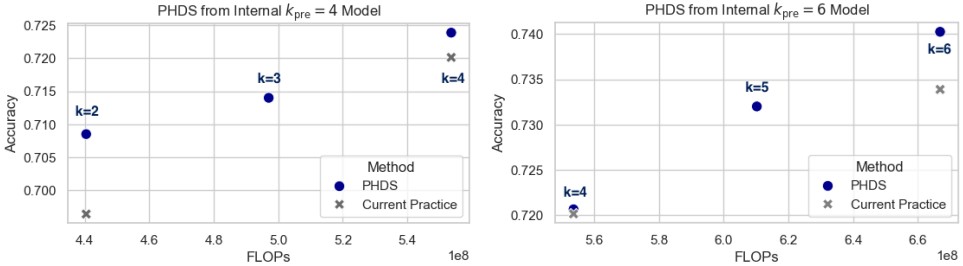

Figure 2: FLOPs vs. average multiple choice accuracy for internal models trained at the current practice (multiple checkpoints for each $k$) (crosses) and PHDS models (dots); all methods are SFTed on `CommonSense170k`. Left: two internal models trained at $k = 2$ and $k = 4$ vs. a single PHDS checkpoint, evaluated at $k = \{2, 3, 4\}$. Right: two internal models trained at $k = 4$ and $k = 6$ vs. a single PHDS checkpoint, evaluated at $k = \{4, 5, 6\}$. PHDS reduced training by 50% compared to current practice and offered support on a wider range of $k$ for each setting.

supported by a single checkpoint. For example, we show that for OLMoE-1B-7B-0125, reducing $k$ at runtime from 8 to 6 decreases multiple choice accuracy by 1.2% (relative) and increases wikitext perplexity by 6.4%; likewise, for Qwen1.5-MoE-A2.7B, reducing $k$ from 4 to 3 results in only a 0.43% accuracy reduction and 1.2% perplexity increase. These findings highlight an overlooked generalization property, with direct implications for efficiency-constrained AI deployment.

Building on this observation, we introduce a lightweight supervised fine tuning (SFT) method to reliably and stably produce models for cross-sparsity deployment. Rather than requiring different checkpoints for different sparsity levels, our proposed method, a **M**ixture **o**f **E**xperts with **P**ost **H**oc **D**eclared **S**parsity (MoE-PHDS), produces a single checkpoint for use across sparsity levels. See figure 1 for an overview and figure 2 for a comparison with current practice. On public models, our method matches or exceeds baselines, and shows benefits, such as higher accuracy and support for extended sparsity ranges, on internal models.

Inference-time sparsity mis-specification can be viewed as a feature of MoE models rather than a bug. Classical mis-specification is when a data-generating regime is not supported by a model class, such as using a linear model for data with a non-linear relationship. Here we study an MoE deployment variant: at inference time we intentionally restrict the model space (fewer experts than used in pretraining), by declaring a smaller $k$ *after* training. PHDS makes this intentional restriction better supported by introducing the model to various $k$ levels during SFT, we keep predictions stable when $k$ is set at runtime. Practically, this yields (*i*) controllability, since a single parameter governs

global sparsity, and (*ii*) predictability with respect to changes in FLOPS or accuracy. In short, we turn "mis-specification" into a serving primitive to produce predictable resource-quality tradeoffs.

From an operational standpoint, global control of sparsity at runtime unlocks capabilities that fixed-sparsity or token-level sparsity models do not have. First, it enables service level agreement (SLA)-aware serving, where a model can adjust sparsity based on user tier, request type, latency budget, or system load. This can mitigate latency spikes and promote tenant fairness without swapping models. Additionally, explicit, discrete $k$ selection per query allows predictable accuracy/latency tradeoffs compared to token-level sparsity. Second, it enables energy-aware inference: sparser models can be used during system wide energy constraints and scale to full model size when resources are not constrained. Finally, it offers substantial operational simplicity: rather than managing a fleet of sparsity-specific models with separate pipelines, drift risks and possibly architectures, practitioners can deploy a single, flexible checkpoint. We focus on small to moderately size models (1B-14.3B), where deployment memory and energy budgets are often the tightest. This paper makes the following contributions:

- **Sparsity level as a serving primitive.** We show that pretrained MoEs tolerate moderate runtime sparsity shifts with minimal loss, challenging the assumption that each sparsity level requires its own checkpoint, allowing sparsity level to act as a serving primitive.
- **MoE-PHDS method.** We propose a lightweight SFT recipe—multi-$k$ training with curriculum anchoring—that enables a single checkpoint to operate flexibly across sparsity levels.
- **Deployment benefits.** We demonstrate that PHDS improves cross-sparsity output agreement (7–22%), yielding stable user-facing behavior while reducing operational complexity: one checkpoint, one control surface, and predictable tradeoffs.

## 2 MoE-PHDS Framework

We introduce an SFT method that allows a pretrained MoE model to support runtime-declared sparsity levels. MoE-PHDS consists of two phases: (1) Multi-$k$ Training, in which the model is fine-tuned by randomly varying the number of active experts $k$ across forward passes, and (2) Curriculum Anchoring, in which training is annealed to a lower $k$ to stabilize expert routing. Once fine-tuned, a single checkpoint can be deployed and reused across a range of sparsity levels. Let $k_{\text{pre}}$ be the sparsity level from the pretrained model, $k_{\text{train}, i}$ the sparsity for forward pass iteration $i$, and $k_{\text{ev}}$ is the evaluation sparsity level. Since expert parameters are only pretrained up to $k_{\text{pre}}$, we restrict training and evaluation to $k_{\text{train},i}, k_{\text{ev}} \leq k_{\text{pre}}$.

### 2.1 Multi-$k$ Training

The goal of Multi-$k$ Training is to expose the model to a range of sparsity levels so that it can generalize beyond fixed $k_{\text{pre}}$. We define a set of candidate sparsity levels, $\mathcal{K}_{\text{train}} \subset \mathbb{N}$ (e.g. $\{4, 5, 6, 7, 8\}$). For each forward pass $i$, we uniformly sample $k_{train, i}$ from $\mathcal{K}_{\text{train}}$ and compute the language modeling loss, such as cross-entropy, under this sparsity. Any auxiliary components (e.g., load-balancing losses, layer norms) are stored and updated per $k$. In practice, we observe rapid convergence, especially when the fine-tuning dataset is distributionally aligned with pretraining data.

### 2.2 Curriculum Anchoring

Although Multi-$k$ Training improves robustness across multiple $k$, performance can degrade at very low $k$ due to interference from higher-$k$ co-activation patterns. To address this, we introduce Curriculum Anchoring: after an initial phase of Multi-$k$ Training, we gradually anneal training toward a fixed, lower $k_{\text{train}}$ (e.g., $k = 2$). This stabilizes expert dynamics at sparse settings and improves reliability when $k_{\text{ev}} \ll k_{\text{pre}}$.

### 2.3 Implementation Details

Pretrained MoE models are not typically designed for runtime sparsity variation. MoE-PHDS introduces minimal modifications to support this flexibility. Let $h \in \mathbb{R}^d$ denote the hidden state,

$W_r \in \mathbb{R}^{d \times E}$ the router weights, and $E$ the number of experts. Router logits are $z = W_r h$, with $p = \text{softmax}(z; k_{\text{pre}})$ the raw gating probabilities. We use a *soft mask* to adapt $p$ to different $k_{\text{train}}$:

$$p_j = \begin{cases} p_j & \text{if } j \in \text{top-k}_{\text{train,i}}(p), \\ \epsilon & \text{if } j \in \text{top-k}_{\text{pre}}(p) \setminus \text{top-k}_{\text{train,i}}(p), \\ 0 & \text{otherwise}, \end{cases} \tag{1}$$

$\epsilon$ is a tunable parameter, we use 1E-6. For unnormalized top-$k$-softmax routers, equation 1 remains unnormalized; for normalized softmax-$k$ routers, the masked probabilities are renormalized. Layer norm parameters and load balancing loss, if applicable, need to be stored and activated by $k_{\text{train}}$.

**Checkpoint Selection.** Multi-$k$ Training with Curriculum Anchoring produces a family of candidate checkpoints. A single checkpoint is chosen after anchoring, either by best validation performance at a target $k_{\text{ev}}$ or by average performance across a range of $k_{\text{ev}}$. We use values at $k_{\text{pre}}$ since we assume it is the default operating point. At runtime, operators declare the desired sparsity level, and the same checkpoint can be reused without retraining.

## 3 EXPERIMENTS

Our experiments test whether one checkpoint can support multiple runtime sparsity levels and when PHDS outperforms oracle or naive baselines. PHDS fine-tuning adds a fraction of pretraining cost, since it is a short SFT pass reusing existing checkpoints. Broadly, public well-tuned models (OL-MoE, Qwen) are already robust to modest sparsity shifts, while less tuned internal models show consistent gains from PHDS. Across models, PHDS maintains $\leq$1–2% relative QA drop when $k$ is reduced by up to 25% (e.g., OLMoE: $k = 8 \rightarrow 6$, Qwen: $k = 4 \rightarrow 3$). Below $k_{\text{pre}}/2$, degradation becomes more pronounced.

### 3.1 EXPERIMENTAL SETUP

#### 3.1.1 PRETRAINED MODELS AND FINE-TUNE DATA

Table 1: Summary of Pretrained Models

| Model | $k_{\text{pre}}$ | Experts | Active Params | Total Params | SFT |
|---|---|---|---|---|---|
| Internal-Baseline-2 | 2 | 16 | 240M | 1.032B | No |
| Internal-Baseline-4 | 4 | 16 | 353M | 1.032B | No |
| Internal-Baseline-6 | 6 | 16 | 466M | 1.032B | No |
| OLMoE-1B-7B-0125 | 8 | 64 | 1B | 7B | No |
| OLMoE-1B-7B-0125-Instruct | 8 | 64 | 1B | 7B | Yes |
| Qwen1.5-MoE-A2.7B | 4 | 60 | 2.7B | 14.3B | No |
| Qwen1.5-MoE-A2.7B-Chat | 4 | 60 | 2.7B | 14.3B | Yes |

**Internal Baseline Model.** We pretrain three MoEs (24 layers; model dim 1024; FFN dim 12,288; 16 experts/layer; 1.032B total params) with $k_{\text{pre}} \in \{2, 4, 6\}$ and softmax–top-$k$ routers. See table 1 for pretrained model specifications.

**OLMoE.** We evaluate OLMoE-1B-7B-0125 and its instruction-tuned variant (Muennighoff et al., 2024). Both use 8/64 experts per layer with a top-$k$–softmax router.

**Qwen.** We evaluate Qwen1.5-MoE-A2.7B and the chat-tuned variant (Qwen Team, 2024), which use 4/60 experts per layer with a top-$k$–softmax router.

**Fine-Tuning Data.** We use `CommonSense170K` from LLM-Adapters (Hu et al., 2023), the `tulu3-sft-olmo-2-mixture` dataset (Lambert et al., 2024), and a high-quality internal mixture ("Internal Baseline Data Set" comprised of licensed; curated public/open data; and a web crawled subset) for additional SFT.

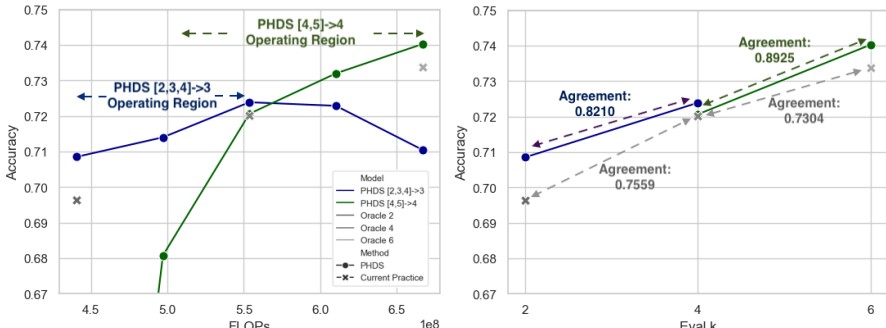

Figure 3: (Left) Average accuracy vs. flops with 4096 context length for Internal Baseline Models. Current practice (Oracle models operating at $k_{\text{pre}}$; denoted by crosses) is compared with PHDS (dots), which produces a range of operating points PHDS operating regions are based on pretrained models, but show ability to meet or outperform baseline models with reduced training costs. (Right) Average accuracy vs. $k_{\text{ev}}$, with answer agreement between checkpoints. Despite similar accuracy, PHDS has increased agreement compared to current practice.

### 3.1.2 FINE-TUNING REGIMES

Prior work on MoE flexibility focuses on token-level adaptability given a fixed global sparsity budget or uses architectural changes. We are interested in MoE robustness to train/inference mis-specification for a fixed architecture. Hence, we use well-specified oracle models as a baseline and compare a set of fine tuning regimes.

**Oracle.** Fine-tune at $k_{\text{pre}}$; denoted Oracle $k_{\text{pre}}$; evaluated at any $k_{\text{ev}}$. This is a well-specified baseline when evaluated at $k_{\text{ev}} = k_{\text{pre}}$.

**Naive.** Fine-tuned at a single, lower $k_{\text{train}} < k_{\text{pre}}$; denoted $k_{\text{pre}} \rightarrow k_{\text{train}}$, e.g. $4 \rightarrow 2$.

**MoE-PHDS.** Sample $k$ uniformly from $\mathcal{K}_{\text{train}}$ during SFT, optionally anneal to a low anchor $k_{\text{train}}$; denoted $\mathcal{K}_{\text{train}} \rightarrow k_{\text{train}}$, e.g., $[2, 3, 4] \rightarrow 2$ or $[2, 3, 4]$ for a non-curriculum trained variant.

### 3.1.3 SELECTION AND EVALUATION PROTOCOL

Unless otherwise noted, we select the checkpoint with the best multiple-choice QA accuracy at $k_{\text{ev}} = k_{\text{pre}}$ after a 5,000-step burn-in as pretrain model-task misalignment may cause accuracy decreases. Public models use `tulu3-sft-olmo-2-mixture` (Tülu-3) for SFT; internal models use the Internal Baseline Data Set or `CommonSense170K`. All regimes are SFTed with the same settings per ablation. We evaluate with `lm-evaluation-harness` for zero shot multiple choice QA: ARC Challenge and Easy (Clark et al., 2018), BoolQ (Clark et al., 2019), HellaSwag (Zellers et al., 2019), PIQA (Bisk et al., 2020), SciQ (Welbl et al., 2017), and Winogrande (Sakaguchi et al., 2021); TriviaQA-fixed (Joshi et al., 2017) for generative QA; and WikiText (Merity et al., 2016) perplexity. For `CommonSense170K` on Internal Baseline Models, we follow LLM-Adapters' stricter matching protocol and evaluate on their multiple choice QA set: ARC-C/E, BoolQ, PIQA, HellaSwag, Winogrande, SocialIQA (Sap et al., 2019), and Openbook QA (Mihaylov et al., 2018). All methods are evaluated at a single checkpoint. For all tables, best results are in **bold**, second-best _underlined_; well-specified models are in blue.

### 3.2 INTERNAL BASELINES: COMMONSENSE170K

We fine-tune untuned Internal Baseline Models on `CommonSense170K` to study (1) accuracy under evaluation sparsity mis-specification, (2) sensitivity of MoE-PHDS to $\mathcal{K}_{\text{train}}$ and curricula, and (3) cross-$k$ agreement. We begin curriculum after 10,000 steps ($\approx$93.9% of epoch 1). Evaluation uses strict matching on 1,000 samples per task. Figure 3 highlights the difference between PHDS and current methods: using Oracle models at $k_{\text{pre}}$, forcing practitioners to maintain multiple

models. In contrast, a single PHDS checkpoint spans multiple sparsity levels, while maintaining accuracy close to or above Oracle at its $k_{pre}$. Full results are in table 2. We find that all families (Oracle/Naive/PHDS) tolerate modest decreases in $k_{ev}$ with limited accuracy loss; increasing beyond $k_{pre}$ does not recover denser-oracle performance. Evaluating below $k_{pre}/2$ produces poor results. PHDS often matches or slightly exceeds the corresponding Oracle around $k_{pre}$, while producing superior results for mis-specified models.

Table 2: `CommonSense170K` SFT: overall accuracy (average across tasks).

| **Model** | $k_{pre}$ | $k_{ev}$=2 | $k_{ev}$=3 | $k_{ev}$=4 | $k_{ev}$=5 | $k_{ev}$=6 |
|---|---|---|---|---|---|---|
| Oracle 2 | 2 | 0.69638 | 0.66750 | 0.70025 | 0.05750 | 0.00013 |
| Oracle 4 | 4 | 0.66113 | 0.70663 | 0.72013 | 0.72688 | 0.71025 |
| PHDS $k$=[2, 3, 4]$\to$3 | 4 | **0.70850** | *0.71400* | **0.72388** | 0.72288 | 0.71050 |
| Naive 4$\to$2 | 4 | *0.69850* | **0.71863** | 0.70488 | 0.56913 | 0.21150 |
| Oracle 6 | 6 | 0.44350 | 0.66013 | 0.72200 | *0.73163* | *0.73388* |
| PHDS $k$=[4, 5]$\to$4 | 6 | 0.47800 | 0.68075 | 0.72063 | **0.73200** | **0.74025** |
| Naive 6$\to$4 | 6 | 0.48275 | 0.68175 | *0.72275* | 0.72788 | 0.73025 |

For operators to vary sparsity at runtime without altering user experience, outputs should remain consistent across $k_{ev}$. We quantify agreement between two models $M_1, M_2$ by averaging discrete answer, $M_{j,i}$, equality over items $i$: $\mathcal{A}(M_1, M_2) = \frac{1}{N}\sum_{i=1}^{N}\mathbf{1}\{M_{1,i} = M_{2,i}\}$. We compare answer agreement of two separate well-specified models with different checkpoints vs. a single PHDS checkpoint evaluated at different sparsity levels in figure 3. Across both $(2, 4)$ and $(4, 6)$ comparisons, a single PHDS checkpoint yields $7\%-22\%$ higher cross-$k$ agreement than using two separate well-specified Oracle checkpoints, *at similar accuracy levels*.

### 3.2.1 INTERNAL BASELINE MODEL: INTERNAL DATA MIXTURE

Here we SFT the Internal Baseline models on a subset of the Internal Data Mixture to measure responsiveness to fine-tuning under mis-specification. Oracle models change little from the pre-trained models, while Naive and PHDS shift their performance profiles. Internal models use $k_{pre} \in \{2, 4, 6\}$. From $k_{pre}$=4: PHDS $[2, 3, 4] \to 3$ and Naive $4 \to 2$. From $k_{pre}$=6: PHDS $[4, 5] \to 4$ and Naive $6 \to 4$. We report multiple-choice QA averages, TriviaQA-fixed (generative QA), and WikiText perplexity (table 3). We find that PHDS is broadly competitive. Across $k_{ev} \in \{2, 3, 4, 5, 6\}$, PHDS is typically within $1\sim2\%$ of the best Oracle for well-specified $k_{ev}$, and often second-best across settings. For generative QA and perplexity, trends mirror QA accuracy: PHDS and Naive are competitive at their target $k$, with Oracle best at its native $k$; PHDS avoids steep degradations when $k_{ev}$ shifts.

### 3.3 OLMoE

We assess mis-specification on OLMoE-1B-7B-0125 with SFT on Tülu-3. We compare Oracle $k$=8, PHDS $[4, 5, 6, 7, 8] \to 5$, PHDS $[4, 5, 6, 7, 8]$, and Naive $8 \to 4$. Results for average multiple choice (Avg), TriviaQA (triv), and wikitext perplexity (wiki) are in table 4; full results by task are in the Appendix in section A.4. We find that: (*i*) all SFT methods are similar, and () robustness to sparsity. At $k_{ev} \in \{8, 7, 6, 5, 4\}$, both accuracy and perplexity remain close across methods. For the base OLMoE model, perplexity stays within $\sim1.5$–$6.4\%$ of the well-specified model for $k_{ev} \in \{6, 7, 8\}$, with overall relative accuracy differences $\leq 1.5\%$.

### 3.4 QWEN

We SFT Qwen1.5-MoE-A2.7B-Chat on Tülu-3, comparing Oracle $k$=4, PHDS $[2, 3, 4] \to 2$, PHDS $[2, 3, 4]$ without curriculum, and Naive $4 \to 2$. Results from the chat-tuned model are in table 5; results for Qwen1.5-MoE-A2.7B are in the Appendix in table 12. We find that all methods remain strong under mis-specification. Accuracy deltas across $k_{ev} \in \{4, 3, 2\}$ are small for both chat and base models; relative accuracy and perplexity degradation from $k_{ev} = 4$ are between 0.2%

Table 3: Internal Baselines fine-tuned on the Internal Baseline Data Set. Models are grouped by $k_{\text{pre}}$ and $k_{\text{ev}}$.

| Method | ARC-C | ARC-E | boolq | hella | piqa | sciq | wino | Avg | triv | wiki |
|---|---|---|---|---|---|---|---|---|---|---|
| | | | | $k_{\text{ev}} = 2$ | | | | | | |
| Oracle 2 | **0.3242** | **0.6650** | **0.6291** | **0.4464** | **0.7323** | **0.899** | **0.5912** | **0.6125** | **0.0813** | **14.039** |
| Oracle 4 | _0.2944_ | 0.5867 | 0.5000 | 0.4235 | 0.7155 | 0.844 | 0.5406 | 0.5578 | 0.0243 | 20.800 |
| [2,3,4]→3 | 0.2833 | _0.6284_ | _0.5654_ | _0.4302_ | **0.7329** | _0.882_ | _0.5533_ | _0.5822_ | _0.0351_ | _16.274_ |
| 4→2 | **0.2952** | **0.6402** | **0.5966** | **0.4379** | _0.7263_ | **0.892** | **0.5943** | **0.5975** | **0.0480** | **15.088** |
| | | | | $k_{\text{ev}} = 3$ | | | | | | |
| Oracle 4 | _0.3250_ | _0.6599_ | 0.6174 | **0.4613** | **0.7459** | 0.901 | _0.5959_ | _0.6157_ | _0.0522_ | _14.184_ |
| [2,3,4]→3 | **0.3251** | **0.6734** | _0.6302_ | _0.4522_ | _0.7448_ | **0.914** | **0.6030** | **0.6204** | **0.0555** | **13.933** |
| 4→2 | 0.3174 | 0.6591 | **0.6398** | 0.4369 | 0.7285 | _0.906_ | 0.5927 | 0.6115 | 0.0480 | 14.889 |
| Oracle 6 | 0.2696 | 0.5749 | 0.5232 | 0.4226 | 0.6882 | 0.799 | 0.5351 | 0.5447 | 0.0269 | 26.144 |
| [4,5]→4 | _0.2858_ | _0.6141_ | _0.5746_ | _0.4376_ | _0.7133_ | _0.851_ | **0.5525** | _0.5755_ | _0.0366_ | _16.848_ |
| 6→4 | **0.2969** | **0.6178** | **0.5780** | **0.4408** | **0.7160** | **0.857** | _0.5446_ | **0.5787** | **0.0442** | **16.762** |
| | | | | $k_{\text{ev}} = 4$ | | | | | | |
| Oracle 4 | **0.3430** | _0.6772_ | 0.6388 | **0.4654** | **0.7465** | **0.910** | **0.6109** | **0.6274** | _0.0556_ | **13.355** |
| [2,3,4]→3 | _0.3353_ | **0.6776** | _0.6440_ | _0.4587_ | _0.7443_ | **0.910** | _0.6054_ | _0.6250_ | **0.0611** | _13.564_ |
| 4→2 | 0.3038 | 0.6595 | **0.6514** | 0.4242 | 0.7171 | 0.909 | 0.5991 | 0.6091 | 0.0441 | 15.972 |
| Oracle 6 | _0.3106_ | 0.6460 | 0.5841 | **0.4633** | 0.7252 | 0.867 | 0.5951 | 0.5988 | _0.0617_ | 15.688 |
| [4,5]→4 | 0.3012 | **0.6670** | _0.6171_ | _0.4609_ | _0.7350_ | _0.886_ | _0.5967_ | _0.6077_ | 0.0591 | _14.024_ |
| 6→4 | **0.3114** | _0.6557_ | **0.6294** | 0.4587 | **0.7367** | **0.887** | **0.6101** | **0.6127** | **0.0636** | **13.955** |
| | | | | $k_{\text{ev}} = 5$ | | | | | | |
| Oracle 6 | **0.3379** | 0.6662 | 0.6205 | **0.4772** | **0.7432** | 0.892 | _0.6069_ | _0.6206_ | **0.0769** | _13.494_ |
| [4,5]→4 | _0.3328_ | **0.6751** | 0.6443 | _0.4680_ | _0.7405_ | _0.893_ | **0.6085** | **0.6232** | _0.0729_ | **13.303** |
| 6→4 | 0.3234 | _0.6709_ | **0.6443** | 0.4592 | 0.7345 | **0.898** | 0.6062 | 0.6195 | 0.0681 | 13.746 |
| | | | | $k_{\text{ev}} = 6$ | | | | | | |
| Oracle 6 | **0.3353** | _0.6793_ | 0.6269 | **0.4761** | **0.7416** | 0.899 | **0.6148** | **0.6247** | **0.0751** | **13.092** |
| [4,5]→4 | _0.3259_ | **0.6806** | _0.6382_ | _0.4665_ | _0.7405_ | _0.900_ | _0.6062_ | _0.6226_ | _0.0673_ | _13.224_ |
| 6→4 | 0.3063 | 0.6667 | **0.6413** | 0.4508 | 0.7334 | **0.902** | 0.6046 | 0.6150 | 0.0605 | 14.219 |

to 0.8% and 1.5% to 1.6% for chat-tuned models on $k_{\text{ev}} = 3$; 2.3% to 2.6% and 6.1% to 7.2% for $k_{\text{ev}} = 2$; for untuned models, 0.2% to 0.4% and 1.2% to 1.3% for $k_{\text{ev}} = 3$; 1.6% to 2.0% and 5.7% to 5.8% for $k_{\text{ev}} = 2$. PHDS often yields small perplexity gains on the chat-tuned variant while matching QA accuracy. Interestingly, TriviaQA-fixed accuracy can increase at reduced $k_{\text{ev}}$.

### 3.5 FIT MECHANISMS AT INCREASED SPARSITY

To understand how and where SFT allows checkpoints to support multiple sparsity levels, we SFTed OLMoE-1B-7B-0125 on `CommonSense170k` and evaluated on the multiple choice evaluation set. This data is somewhat out of distribution; we varied free parameters during fit with models *Baseline*, *Gate*, *Expert*, *Attention*, and *Expert and Gate* with Oracle and MoE-PHDS. As $k_{\text{ev}}$ decreases, attention carries more lift than expert-only refits; at $k_{\text{ev}} = k_{\text{pre}}$, expert refits dominate. Results are in figure 3.5. Full protocol and plots appear in Appendix A.6.

## 4 RELATED WORK

**Many-in-One Models.** Prior work enables a single network to run across compute budgets via global, pretraining-time sparsity or nested subnetworks. Slimmable and Once-for-All train

Table 4: OLMoE fine-tuned on Tülu-3.

| | | OLMoE-1B-7B-0125 | | | OLMoE-0125-1B-7B-Instruct | | |
|---|---|---|---|---|---|---|---|
| Method | $k_{ev}$ | Avg | triv | wiki | Avg | triv | wiki |
| Oracle | 8 | *0.7050* | *0.4889* | 9.407 | *0.7067* | 0.3977 | 15.810 |
| [4,5,6,7,8]→5 | 8 | 0.6997 | 0.4585 | *8.923* | **0.7068** | *0.4131* | 15.742 |
| [4,5,6,7,8] | 8 | **0.7051** | **0.4901** | 8.903 | 0.7041 | 0.4119 | *15.484* |
| 8→4 | 8 | 0.7021 | 0.4794 | 9.485 | 0.7059 | **0.4174** | **15.488** |
| Oracle | 7 | **0.7017** | **0.4928** | 9.557 | **0.7022** | 0.3902 | 16.238 |
| [4,5,6,7,8]→5 | 7 | 0.6939 | 0.4578 | *9.075* | 0.7006 | 0.3928 | 16.280 |
| [4,5,6,7,8] | 7 | *0.7001* | *0.4815* | **9.056** | 0.6999 | *0.3973* | **16.014** |
| 8→4 | 7 | 0.6966 | 0.4793 | 9.639 | *0.7008* | **0.4003** | *16.042* |
| Oracle | 6 | **0.6967** | **0.4817** | 9.985 | **0.6953** | 0.3756 | 17.372 |
| [4,5,6,7,8]→5 | 6 | 0.6893 | 0.4667 | *9.496* | 0.6934 | 0.3842 | 17.426 |
| [4,5,6,7,8] | 6 | *0.6966* | *0.4767* | **9.474** | 0.6906 | *0.3862* | **17.148** |
| 8→4 | 6 | 0.6924 | 0.4737 | 10.091 | *0.6943* | **0.3909** | *17.188* |
| Oracle | 5 | **0.6849** | **0.4487** | 10.975 | *0.6837* | 0.3686 | 19.687 |
| [4,5,6,7,8]→5 | 5 | 0.6742 | 0.4242 | *10.431* | 0.6831 | **0.3745** | 19.775 |
| [4,5,6,7,8] | 5 | *0.6822* | 0.4293 | **10.408** | 0.6793 | 0.3695 | **19.534** |
| 8→4 | 5 | 0.6806 | *0.4448* | 11.151 | **0.6847** | *0.3700* | *19.621* |
| Oracle | 4 | *0.6646* | *0.3787* | 13.032 | *0.6627* | 0.3304 | **24.574** |
| [4,5,6,7,8]→5 | 4 | 0.6590 | 0.3605 | *12.357* | **0.6650** | **0.3339** | 24.921 |
| [4,5,6,7,8] | 4 | **0.6668** | 0.3775 | **12.304** | 0.6580 | 0.3305 | *24.640* |
| 8→4 | 4 | 0.6587 | **0.3886** | 13.424 | 0.6603 | *0.3316* | 24.965 |

Table 5: Qwen fine-tuned on Tülu-3.

| | | Qwen1.5-MoE-A2.7B-Chat | | | Qwen1.5-MoE-A2.7B | | |
|---|---|---|---|---|---|---|---|
| Method | $k_{ev}$ | Avg | triv | wiki | Avg | triv | wiki |
| Oracle | 4 | 0.7001 | 0.0563 | 11.436 | 0.7069 | **0.0287** | **10.223** |
| [2,3,4]→2 | 4 | 0.7005 | *0.0585* | *11.373* | **0.7073** | 0.0271 | 10.244 |
| [2,3,4] | 4 | **0.7022** | **0.0631** | **11.331** | 0.7059 | 0.0269 | 10.227 |
| 4→2 | 4 | *0.7007* | 0.0526 | 11.420 | *0.7072* | *0.0284* | *10.225* |
| Oracle | 3 | 0.6972 | 0.0633 | 11.608 | *0.7050* | *0.0337* | *10.355* |
| [2,3,4]→2 | 3 | *0.6979* | 0.0642 | *11.549* | 0.7043 | 0.0329 | 10.367 |
| [2,3,4] | 3 | 0.6968 | **0.0664** | **11.502** | 0.7036 | 0.0317 | 10.364 |
| 4→2 | 3 | **0.6992** | *0.0661* | 11.605 | **0.7061** | **0.0342** | **10.353** |
| Oracle | 2 | 0.6821 | 0.0625 | 12.225 | **0.6952** | 0.0334 | **10.810** |
| [2,3,4]→2 | 2 | *0.6845* | *0.0662* | *12.070* | 0.6929 | **0.0342** | 10.832 |
| [2,3,4] | 2 | **0.6854** | **0.0675** | **12.033** | 0.6928 | 0.0317 | 10.821 |
| 4→2 | 2 | 0.6835 | 0.0643 | 12.132 | *0.6931* | *0.0341* | *10.811* |

width/subnet sets for CNNs (Yu et al., 2019; Yu & Huang, 2019; Li et al., 2021; Cai et al., 2020; Lou et al., 2021); transformer variants drop tokens or learn nested blocks (DynamicViT, ViT-Slimmable, Matryoshka, MatFormer) (Rao et al., 2021; Yin et al., 2022; Kusupati et al., 2022; Devvrit et al., 2024). For LLMs, Flextron fits routers post-training (Cai et al., 2024); pruning methods (e.g., retraining-free/Fisher) remove heads or filters (Kwon et al., 2022). These methods span multiple operating points but typically require bespoke pretraining, architecture changes, or storing multiple subnetworks.

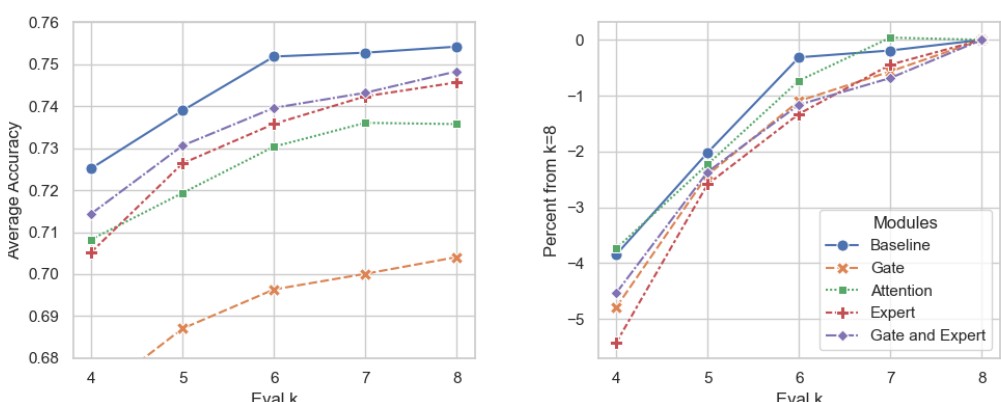

Figure 4: Average multiple choice QA accuracy vs. $k_{\text{ev}}$ for parameter-subset refits for OLMoE-1B-7B-0125 on `CommonSense170k` with Oracle (left: average accuracy; right: relative accuracy reduction per parameter subset vs. $k_{\text{ev}} = 8$).

**Sparse MoEs.** Sparse MoEs are sparse models by design: a global sparsity parameter determines how many experts are active per token (Fedus et al., 2022; Riquelme et al., 2021). Subsequent work has explored richer forms of token-level sparsity. Token-aware schemes include probabilistic top-$k(P)$ gating (Huang et al., 2024), dynamic routing (Alizadeh-Vahid et al., 2024), the addition of null experts (Zeng et al., 2024; Team et al., 2025), and multiplier layers for experts in TC-Experts (Yan et al., 2025). Posttraining methods such as DynaMoE (Nishu et al., 2025) convert dense LLMs into token-sparse adaptive MoEs. Token-adaptive MoEs spend a fixed global budget more efficiently; PHDS changes the global budget at runtime. These approaches compose.

## 5 DISCUSSION

In this paper, we (*i*) showed that pretrained sparse MoE models are more robust to runtime changes in sparsity than commonly assumed, (*ii*) demonstrated that sparsity can be an MoE serving primitive from a single checkpoint, and (*iii*) introduced MoE-PHDS, which allows practitioners to use SFT to make their existing models more robust to sparsity mis-specification. While naive SFT often works, MoE-PHDS provides added benefits for less tuned models and extends support across a slightly larger range of $k_{\text{ev}}$. In practice, operators can often safely reduce $k$ by 20–30% with minimal loss, while larger reductions should be treated as best-effort. Although we evaluate moderate-scale MoEs, PHDS is most valuable where latency, energy, or memory are tight. Our experiments span models from 1B–14.3B parameters, a regime where memory and energy budgets are tightest. In these settings, multiple checkpoints are impractical and token-level adaptivity introduces variance, whereas a single global sparsity knob offers predictable accuracy–efficiency trade-offs without architectural changes. Higher cross-$k$ agreement further preserves the model's "feel" as sparsity varies at runtime.

**Limitations.** Results are from smaller models; scalability to larger ones is unknown. We study routed, equal-sized experts only; partial routing or heterogeneous expert sizes may behave differently. We also omit generation-heavy tasks (coding, summarization) and answer-style analyses, so some gains may reflect stylistic shifts rather than ability. Reported safe ranges are model and task dependent and need further study.

**Reproducibility.** When possible, we used public models; we trained and evaluated on public data sets with standard harnesses. Diffs from OLMoE-1B-7B-0125 and Qwen1.5-MoE-A2.7B to support PHDS will be available in a git repo pending institutional approval, along with a `.json` file with experimental settings.

**LLM Usage.** LLMs were used in this work for outlining, text editing, and literature search.

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

## A  APPENDIX

### A.1  LITERATURE COMPARISON

**Positioning.** Literature is summarized in table 6. Prior work demonstrates that both dense and sparse models can be adapted to multiple computational settings, but with important limitations: dense models typically rely on pretraining across subnetworks or block structures, while MoEs fix a global sparsity parameter in advance. Token- or layer-level dynamic gating can be a powerful way to optimize how computational budget is spent for a fixed global sparsity level, but it injects variance (latency fluctuates with content) and adds policy complexity with additional tunable parameters. In contrast, to our knowledge, MoE-PHDS is the first method to enable global runtime sparsity control from a single checkpoint. Our approach only requires lightweight supervised fine tuning, avoids maintaining multiple subnetworks, and directly supports deployment across a set of operating points

due to predictability, simplicity, and composability. Therefore this method complements, rather than competes with, finer-grained adaptivity.

Table 6: Comparison of methods enabling multiple operating points from a single model. MoE-PHDS uniquely supports *global runtime sparsity control* from a single checkpoint.

| Method | Train | Runtime | Sparsity | Family |
|---|---|---|---|---|
| *MoE (ours)* | | | | |
| MoE-PHDS | SFT | Yes (1 checkpoint) | Global ($k$) | MoE |
| *Dense Networks / FFNs* | | | | |
| Slimmable (Yu et al., 2019) | Pre | Yes (width) | Global | CNN/FFN |
| US-Net (Yu & Huang, 2019) | Pre | Yes (width) | Global | CNN/FFN |
| OFA (Cai et al., 2020) | Pre | No | Global | FFN |
| Dyn-OFA (Cai et al., 2020) | Pre | Yes (stored nets) | Global | FFN |
| DS-Net (Lou et al., 2021) | Pre | Yes (stored filt.) | Global | FFN |
| MatFormer (Devvrit et al., 2024) | Pre | Yes (blocks) | Block | Trans. |
| Matryoshka (Kusupati et al., 2022) | Pre | Yes (nested) | Block | Trans. |
| Flextron (Cai et al., 2024) | Post | Yes (router) | Layer | LLM |
| RF-Pruning (Kwon et al., 2022) | Post | No | Head/Fil. | Trans. |
| *Sparse MoEs* | | | | |
| top-$k(P)$ (Huang et al., 2024) | Pre | No | Token | MoE |
| AdaMoE (Zeng et al., 2024) | Pre | No | Token | MoE |
| TC-MoE (Yan et al., 2025) | Pre | No | Token | MoE |
| DynaMoE (Nishu et al., 2025) | Post | No | Token | MoE/LLM |
| LongCat (Team et al., 2025) | Pre | No | Token | MoE |

## A.2 EXPERIMENTAL SETTINGS

We used the settings summarized in table 7 for experiments. We run one seed for SFT (budget-constrained) and reuse that checkpoint across $k_{ev}$; evaluation uses fixed harness seeds. For each we experiment, we ran a set of longer SFT trials to determine the number of tokens seen for mass ablations. Initial experiments were run for Oracle and PHDS settings, with a shortened schedule applied to all other ablations. Truncated schedule size was determined by where best checkpoints were selected in the initial run phases.

PHDS has three main tunble parameters: $\epsilon$, $\mathcal{K}_{\text{train}}$, and Curriculum value. We chose $\epsilon$ by running ablations from 1E-1 to 1E-8; no material difference was seen below 1E-4, so we selected 1E-6 for all experiments. This value is large enough that there is *some* gradient flow and back propagation does not collapse, but small enough that changes when these values are included only have minor effects. The inclusion of a soft mask is also done for deployment ease, as methods like jax work best without changes to array sizes. In general, based on general performance across ablations, we use $\mathcal{K}_{\text{train}} = \{k_{\text{pre}}/2, \ldots, k_{\text{pre}} - 1, k_{\text{pre}}\}$.

Table 7: Experimental settings. LB is load balancing value.

| Experiment | Initial Tokens | Ablation Tokens | GPUs | LB |
|---|---|---|---|---|
| Internal Baseline: CS170k | 491M | 246M | 2xA100-40GB | 0.01 |
| Internal Baseline: Internal Data | 393M | 393M | 8xA100-40GB | 0.01 |
| OLMoE: Tülu-3 | 39B | 7.8B | 8xA100-40GB | 0.0 |
| OLMoE: CS170k | 327M | 327M | 8xA100-40GB | 0.0 |
| Qwen: Tülu-3 | 39B | 7.8B | 8xA100-40GB | 0.0 |

### A.3 Internal Baseline Models: CommonSense170k

**Ablations on MoE-PHDS Parameters.** MoE-PHDS has two main tunable parameters: the sampling set, $\mathcal{K}_{\text{train}}$, and the curriculum training values. In table 8, we train across sampling sets. In table 9, we fix a subset of sampling sets and train across curriculum values.

Table 8: Internal Baseline Models: CommonSense170k SFT, ablations across $\mathcal{K}_{\text{train}}$ sets. Overall accuracy on CommonSense multiple choice. Best values are **bold** and second best _underlined_, grouped by $k_{\text{pre}}$ and $k_{\text{ev}}$. Untargeted $k_{\text{ev}}$ are denoted by $--$.

| **Model** | $k_{\text{pre}}$ | $k_{\text{ev}} = 2$ | $k_{\text{ev}} = 3$ | $k_{\text{ev}} = 4$ | $k_{\text{ev}} = 5$ | $k_{\text{ev}} = 6$ |
|---|---|---|---|---|---|---|
| PHDS k=[2,3] | 4 | 0.677000 | 0.708500 | 0.715625 | – | – |
| PHDS k=[2,3] $\to$ 2 | 4 | 0.682875 | 0.713875 | **0.720500** | – | – |
| PHDS k=[2,3,4] | 4 | 0.678750 | 0.712000 | 0.718000 | – | – |
| PHDS k=[2,3,4] $\to$ 2 | 4 | **0.695000** | **0.716500** | 0.717375 | – | – |
| PHDS k=[2,4] | 4 | 0.666625 | 0.698375 | 0.708750 | – | – |
| PHDS k=[2,4] $\to$ 2 | 4 | _0.686625_ | _0.716125_ | _0.718875_ | – | – |
| PHDS k=[4,5] | 6 | – | 0.683250 | 0.718875 | 0.729500 | 0.733250 |
| PHDS k=[4,5] $\to$ 4 | 6 | – | 0.680750 | _0.720625_ | **0.732000** | **0.740250** |
| PHDS k=[4,5,6] | 6 | – | 0.670250 | 0.706750 | 0.725000 | 0.730125 |
| PHDS k=[4,5,6] $\to$ 4 | 6 | – | _0.692500_ | 0.718250 | _0.731750_ | 0.733625 |
| PHDS k=[3,4,5,6] | 6 | – | **0.694125** | 0.720250 | 0.728625 | 0.729375 |
| PHDS k=[3,4,5,6] $\to$ 4 | 6 | – | 0.689875 | **0.723750** | 0.731375 | _0.734250_ |

In experiments for table 8, we found that using all values between $k_{\text{pre}}/2$ and $k_{\text{pre}}$ produces consistently solid results. For higher $k_{\text{pre}}$ values, non-inclusive subsets between $k_{\text{pre}}/2$ and $k_{\text{pre}}$ work well, and curriculum training consistently increases accuracy. In the experiments for table 9, we found that the best results are consistently from $k$ just above $k_{\text{pre}}/2$, and that results are consistent across $\mathcal{K}_{\text{train}}$ groups based given $k_{\text{pre}}$.

Table 9: CommonSense 170k Fine-tuning: overall accuracy by $\mathcal{K}_{\text{train}}$ and curriculum.

| v **Model** | $k_{\text{pre}}$ | $k_{\text{ev}} = 2$ | $k_{\text{ev}} = 3$ | $k_{\text{ev}} = 4$ | $k_{\text{ev}} = 5$ | $k_{\text{ev}} = 6$ |
|---|---|---|---|---|---|---|
| PHDS k=[2,3,4] | 4 | 0.678750 | 0.712000 | _0.718000_ | – | – |
| PHDS k=[2,3,4] $\to$ 1 | 4 | 0.675125 | 0.700375 | 0.708375 | – | – |
| PHDS k=[2,3,4] $\to$ 2 | 4 | _0.695000_ | **0.716500** | 0.717375 | – | – |
| PHDS k=[2,3,4] $\to$ 3 | 4 | **0.708500** | _0.714000_ | **0.723875** | – | – |
| PHDS k=[2,3,4] $\to$ 4 | 4 | 0.666250 | 0.706750 | 0.714625 | – | – |
| PHDS k=[4,5] | 6 | – | **0.683250** | _0.718875_ | _0.729500_ | 0.733250 |
| PHDS k=[4,5] $\to$ 2 | 6 | – | 0.679500 | 0.713750 | 0.717875 | 0.723125 |
| PHDS k=[4,5] $\to$ 3 | 6 | – | 0.668750 | 0.704250 | 0.713875 | 0.715625 |
| PHDS k=[4,5] $\to$ 4 | 6 | – | _0.680750_ | **0.720625** | **0.732000** | **0.740250** |
| PHDS k=[4,5] $\to$ 5 | 6 | – | 0.676875 | 0.716250 | 0.728875 | _0.733375_ |
| PHDS k=[4,5] $\to$ 6 | 6 | – | 0.574500 | 0.622750 | 0.729000 | 0.733250 |
| PHDS k=[3,4,5,6] | 6 | – | _0.694125_ | _0.720250_ | 0.728625 | 0.729375 |
| PHDS k=[3,4,5,6] $\to$ 2 | 6 | – | 0.677500 | 0.706750 | 0.718375 | 0.721125 |
| PHDS k=[3,4,5,6] $\to$ 3 | 6 | – | **0.695125** | 0.704250 | 0.724625 | 0.722500 |
| PHDS k=[3,4,5,6] $\to$ 4 | 6 | – | 0.689875 | **0.723750** | **0.731375** | **0.734250** |
| PHDS k=[3,4,5,6] $\to$ 5 | 6 | – | 0.680750 | 0.711875 | 0.723000 | 0.725125 |
| PHDS k=[3,4,5,6] $\to$ 6 | 6 | – | 0.574500 | 0.719875 | _0.728875_ | _0.732875_ |

### A.4 OLMoE Experiments

Full results from OLMoE on Tülu-3 is given in tables 10 and 11.

Table 10: OLMoE fine-tuned on Tülu-3. Best per block in **bold**, second-best _underlined_. Well-specified methods are in blue.

| **OLMoE-1B-7B-0125-Instruct** | | | | | | | | | | |
|---|---|---|---|---|---|---|---|---|---|---|
| **Model** | **ARC-C** | **ARC-E** | **boolq** | **hella** | **piqa** | **sciq** | **wino** | **Avg** | **triv** | **wiki** |
| $k_{\mathrm{ev}} = 8$ | | | | | | | | | | |
| Oracle | 0.4642 | **0.7412** | 0.7563 | **0.5977** | _0.7622_ | _0.949_ | **0.6764** | _0.7067_ | 0.3977 | 15.810 |
| [4,...,8]→5 | **0.4701** | 0.7336 | **0.7615** | _0.5950_ | 0.7590 | **0.956** | _0.6725_ | **0.7068** | _0.4131_ | 15.742 |
| [4,5,6,7,8] | 0.4642 | 0.7340 | 0.7596 | 0.5932 | **0.7644** | 0.948 | 0.6654 | 0.7041 | 0.4119 | _15.484_ |
| 8→4 | **0.4701** | _0.7370_ | _0.7599_ | 0.5941 | 0.7617 | _0.949_ | 0.6693 | 0.7059 | **0.4174** | **15.488** |
| $k_{\mathrm{ev}} = 7$ | | | | | | | | | | |
| Oracle | 0.4582 | **0.7391** | 0.7563 | 0.5922 | 0.7601 | _0.950_ | **0.6598** | **0.7022** | 0.3902 | 16.238 |
| [4,...,8]→5 | 0.4582 | 0.7294 | 0.7563 | **0.5945** | _0.7606_ | 0.948 | 0.6575 | 0.7006 | 0.3928 | 16.280 |
| [4,5,6,7,8] | _0.4608_ | 0.7231 | **0.7575** | _0.5928_ | 0.7552 | **0.951** | _0.6590_ | 0.6999 | _0.3973_ | **16.014** |
| 8→4 | **0.4633** | _0.7298_ | **0.7575** | 0.5909 | **0.7617** | 0.949 | 0.6535 | _0.7008_ | **0.4003** | _16.042_ |
| $k_{\mathrm{ev}} = 6$ | | | | | | | | | | |
| Oracle | **0.4471** | _0.7201_ | _0.7609_ | **0.5895** | **0.7590** | 0.947 | _0.6433_ | **0.6953** | 0.3756 | 17.372 |
| [4,...,8]→5 | _0.4437_ | 0.7142 | **0.7667** | _0.5884_ | 0.7514 | **0.951** | 0.6385 | 0.6934 | 0.3842 | 17.426 |
| [4,5,6,7,8] | 0.4352 | 0.7130 | 0.7590 | 0.5878 | 0.7503 | **0.951** | 0.6377 | 0.6906 | _0.3862_ | **17.148** |
| 8→4 | 0.4428 | **0.7205** | 0.7606 | 0.5871 | _0.7563_ | 0.947 | **0.6464** | _0.6943_ | **0.3909** | _17.188_ |
| $k_{\mathrm{ev}} = 5$ | | | | | | | | | | |
| Oracle | _0.4334_ | 0.7029 | 0.7413 | **0.5774** | 0.7443 | _0.947_ | _0.6393_ | _0.6837_ | 0.3686 | 19.687 |
| [4,...,8]→5 | 0.4300 | **0.7054** | **0.7468** | 0.5742 | 0.7437 | 0.943 | 0.6385 | 0.6831 | **0.3745** | 19.775 |
| [4,5,6,7,8] | 0.4232 | 0.6944 | _0.7419_ | _0.5743_ | **0.7508** | 0.946 | 0.6243 | 0.6793 | 0.3695 | **19.534** |
| 8→4 | **0.4377** | _0.7050_ | _0.7419_ | 0.5721 | _0.7481_ | **0.948** | **0.6401** | **0.6847** | _0.3700_ | _19.621_ |
| $k_{\mathrm{ev}} = 4$ | | | | | | | | | | |
| Oracle | 0.3805 | **0.6894** | **0.7287** | **0.5543** | 0.7296 | **0.942** | 0.6140 | _0.6627_ | 0.3304 | **24.574** |
| [4,...,8]→5 | **0.3899** | _0.6814_ | _0.7278_ | _0.5509_ | **0.7394** | _0.939_ | **0.6267** | **0.6650** | **0.3339** | 24.921 |
| [4,5,6,7,8] | 0.3823 | 0.6692 | 0.7242 | 0.5472 | _0.7329_ | 0.937 | 0.6133 | 0.6580 | 0.3305 | _24.640_ |
| 8→4 | _0.3874_ | 0.6768 | 0.7196 | 0.5490 | 0.7301 | 0.938 | _0.6212_ | 0.6603 | _0.3316_ | 24.965 |

## A.5 QWEN

Full results for Qwen on Tülu-3 are given in table 12.

## A.6 MECHANISMS OF ROBUSTNESS AT REDUCED $k$

**Setup.** On OLMoE-1B-7B-0125, we SFT on CommonSense170K with subset refits: *Baseline*, *Gate*, *Expert*, *Attention*, *Expert and Gate*, under Oracle and PHDS [4,5,6,7,8]→5 regimes. Curriculum scheduling is introduced to PHDS after 93.9% of epoch 1; all runs are done through two full epochs. Checkpoints are selected by best MC-QA accuracy at $k_{\mathrm{ev}}=8$.

**Metrics.** Overall MC-QA accuracy and relative drop vs. accuracy for $k_{\mathrm{ev}} = 8$ by parameter subset. Metrics are reported by parameter subset to understand which subsets have less relative degradation at low $k_{\mathrm{ev}}$, even if they have poorer fits at $k_{\mathrm{ev}} = 8$.

**Findings.** Our results with MoE-PHDS are similar to those for Oracle, with *Expert* adding the majority of fit value at $k_{\mathrm{ev}} = 8$, but with *Attention* contributing significant value at $k_{\mathrm{ev}} = 4$.

Table 11: OLMoE fine-tuned on Tülu-3. Best per block in **bold**, second-best _underlined_. Well-specified methods are in blue.

| **OLMoE-1B-7B-0125** | | | | | | | | | | |
|---|---|---|---|---|---|---|---|---|---|---|
| **Model** | **ARC-C** | **ARC-E** | **boolq** | **hella** | **piqa** | **sciq** | **wino** | **Avg** | **triv** | **wiki** |
| $k_{ev} = 8$ | | | | | | | | | | |
| Oracle | _0.4582_ | **0.7757** | 0.7040 | 0.5673 | 0.7797 | **0.955** | _0.6953_ | _0.7050_ | _0.4889_ | 9.407 |
| [4,...,8]→5 | 0.4471 | 0.7605 | 0.6997 | _0.5742_ | _0.7856_ | 0.932 | **0.6985** | 0.6997 | 0.4585 | _8.923_ |
| [4,5,6,7,8] | **0.4659** | _0.7694_ | **0.7070** | **0.5764** | _0.7856_ | 0.937 | 0.6946 | **0.7051** | **0.4901** | **8.903** |
| 8→4 | _0.4573_ | 0.7668 | _0.7043_ | 0.5635 | **0.7862** | _0.943_ | 0.6938 | 0.7021 | 0.4794 | 9.485 |
| $k_{ev} = 7$ | | | | | | | | | | |
| Oracle | 0.4497 | **0.7723** | _0.7018_ | 0.5680 | **0.7835** | **0.951** | **0.6859** | **0.7017** | **0.4928** | 9.557 |
| [4,...,8]→5 | 0.4428 | 0.7508 | 0.6899 | _0.5747_ | **0.7835** | 0.932 | 0.6835 | 0.6939 | 0.4578 | _9.075_ |
| [4,5,6,7,8] | _0.4531_ | _0.7626_ | **0.7061** | 0.5757 | **0.7835** | 0.934 | **0.6859** | _0.7001_ | _0.4815_ | **9.056** |
| 8→4 | **0.4565** | 0.7597 | 0.6896 | 0.5666 | 0.7786 | _0.944_ | 0.6811 | 0.6966 | 0.4793 | 9.639 |
| $k_{ev} = 6$ | | | | | | | | | | |
| Oracle | 0.4403 | **0.7677** | **0.7009** | 0.5665 | 0.7780 | **0.952** | 0.6717 | **0.6967** | **0.4817** | 9.985 |
| [4,...,8]→5 | _0.4471_ | 0.7479 | 0.6865 | _0.5703_ | 0.7780 | 0.928 | 0.6669 | 0.6893 | 0.4667 | _9.496_ |
| [4,5,6,7,8] | **0.4565** | _0.7534_ | _0.6994_ | 0.5726 | _0.7840_ | 0.934 | **0.6764** | _0.6966_ | _0.4767_ | **9.474** |
| 8→4 | 0.4445 | 0.7500 | 0.6884 | 0.5637 | **0.7856** | _0.938_ | **0.6764** | 0.6924 | 0.4737 | 10.091 |
| $k_{ev} = 5$ | | | | | | | | | | |
| Oracle | _0.4206_ | **0.7370** | _0.6957_ | 0.5559 | 0.7704 | **0.942** | **0.6725** | **0.6849** | **0.4487** | 10.975 |
| [4,...,8]→5 | 0.4078 | 0.7151 | 0.6841 | _0.5619_ | 0.7629 | _0.934_ | 0.6535 | 0.6742 | 0.4242 | _10.431_ |
| [4,5,6,7,8] | _0.4206_ | 0.7311 | **0.7028** | **0.5630** | 0.7720 | 0.933 | 0.6527 | _0.6822_ | 0.4293 | **10.408** |
| 8→4 | **0.4232** | _0.7323_ | 0.6865 | 0.5540 | _0.7709_ | _0.934_ | _0.6630_ | 0.6806 | _0.4448_ | 11.151 |
| $k_{ev} = 4$ | | | | | | | | | | |
| Oracle | _0.4053_ | **0.7109** | 0.6706 | 0.5386 | 0.7628 | **0.930** | _0.6338_ | _0.6646_ | _0.3787_ | 13.032 |
| [4,...,8]→5 | 0.3984 | 0.6949 | _0.6722_ | _0.5443_ | 0.7584 | 0.920 | 0.6251 | 0.6590 | 0.3605 | _12.357_ |
| [4,5,6,7,8] | **0.4138** | _0.7024_ | **0.6798** | **0.5471** | 0.7661 | _0.927_ | 0.6314 | **0.6668** | 0.3775 | **12.304** |
| 8→4 | 0.3993 | 0.7020 | 0.6648 | 0.5369 | _0.7633_ | 0.909 | **0.6354** | 0.6587 | **0.3886** | 13.424 |

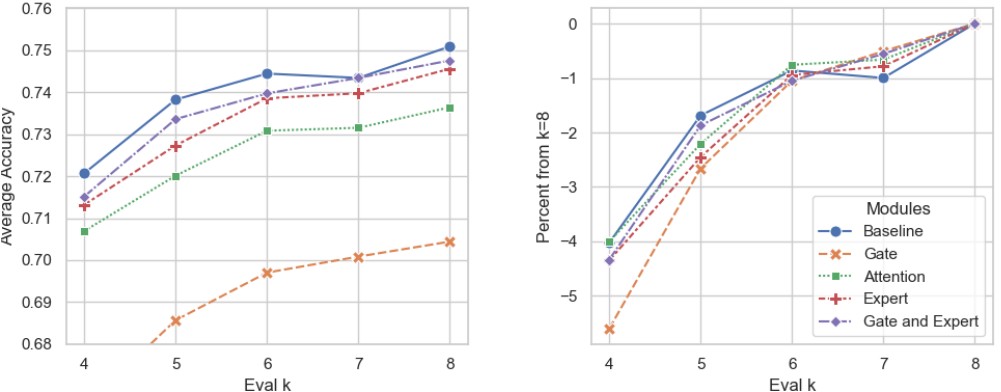

Figure 5: Average multiple choice QA accuracy vs. $k_{ev}$ for parameter-subset refits for OLMoE-1B-7B-0125 on `CommonSense170k` with PHDS [4,5,6,7,8]→5 (left: average accuracy; right: relative drop per parameter subset vs. $k_{ev} = 8$).

Table 12: Qwen: fine-tuned on Tülu-3 for the untuned model. Best per block in **bold**, second-best _underlined_. Well-specified methods are in blue.

**Qwen1.5-MoE-A2.7B**

| Model | ARC-C | ARC-E | boolq | hella | piqa | sciq | wino | Avg | triv | wiki |
|---|---|---|---|---|---|---|---|---|---|---|
| | | | | | $k_{\mathrm{ev}}=4$ | | | | | |
| Oracle | 0.4104 | **0.7302** | _0.7911_ | _0.5806_ | 0.7982 | **0.945** | 0.6906 | 0.7069 | **0.0287** | **10.223** |
| [2,3,4]→2 | **0.4172** | 0.7273 | 0.7884 | 0.5798 | **0.8020** | 0.942 | **0.6946** | **0.7073** | 0.0271 | 10.244 |
| [2,3,4] | 0.4121 | 0.7290 | 0.7859 | 0.5805 | _0.7987_ | 0.942 | _0.6930_ | 0.7059 | 0.0269 | 10.227 |
| 4→2 | **0.4172** | _0.7298_ | **0.7920** | **0.5813** | _0.7987_ | _0.944_ | 0.6875 | _0.7072_ | _0.0284_ | _10.225_ |
| | | | | | $k_{\mathrm{ev}}=3$ | | | | | |
| Oracle | 0.4002 | 0.7281 | 0.7872 | **0.5781** | _0.7976_ | **0.949** | 0.6946 | _0.7050_ | _0.0337_ | _10.355_ |
| [2,3,4]→2 | **0.4044** | 0.7252 | 0.7887 | 0.5759 | 0.7965 | 0.948 | 0.6914 | 0.7043 | 0.0329 | 10.367 |
| [2,3,4] | 0.3985 | _0.7302_ | **0.7905** | 0.5766 | _0.7976_ | 0.947 | 0.6851 | 0.7036 | 0.0317 | 10.364 |
| 4→2 | _0.4019_ | **0.7319** | _0.7893_ | _0.5775_ | **0.7992** | **0.949** | _0.6938_ | **0.7061** | **0.0342** | **10.353** |
| | | | | | $k_{\mathrm{ev}}=2$ | | | | | |
| Oracle | **0.4027** | **0.7146** | **0.7789** | **0.5680** | **0.7938** | **0.946** | 0.6622 | **0.6952** | 0.0334 | **10.810** |
| [2,3,4]→2 | _0.4019_ | 0.7092 | 0.7746 | 0.5658 | 0.7884 | **0.946** | _0.6646_ | 0.6929 | **0.0342** | 10.832 |
| [2,3,4] | 0.3959 | 0.7075 | 0.7774 | 0.5673 | 0.7884 | **0.946** | **0.6669** | 0.6928 | 0.0317 | 10.821 |
| 4→2 | 0.4002 | _0.7113_ | _0.7786_ | **0.5680** | **0.7938** | **0.946** | 0.6535 | _0.6931_ | _0.0341_ | _10.811_ |

**Qwen1.5-MoE-A2.7B-Chat**

| Model | ARC-C | ARC-E | boolq | hella | piqa | sciq | wino | Avg | triv | wiki |
|---|---|---|---|---|---|---|---|---|---|---|
| | | | | | $k_{\mathrm{ev}}=4$ | | | | | |
| Oracle | _0.3985_ | 0.7012 | **0.8089** | 0.5936 | 0.7894 | 0.947 | 0.6622 | 0.7001 | 0.0563 | 11.436 |
| [2,3,4]→2 | **0.4002** | _0.7029_ | _0.8080_ | **0.5950** | 0.7861 | 0.946 | 0.6653 | 0.7005 | _0.0585_ | _11.373_ |
| [2,3,4] | _0.3985_ | _0.7029_ | 0.8061 | _0.5943_ | **0.7927** | **0.949** | **0.6717** | **0.7022** | **0.0631** | **11.331** |
| 4→2 | 0.3951 | **0.7045** | 0.8076 | 0.5936 | _0.7900_ | _0.948_ | _0.6661_ | _0.7007_ | 0.0526 | 11.420 |
| | | | | | $k_{\mathrm{ev}}=3$ | | | | | |
| Oracle | 0.3951 | 0.7003 | 0.8037 | 0.5914 | 0.7840 | **0.949** | _0.6567_ | 0.6972 | 0.0633 | 11.608 |
| [2,3,4]→2 | _0.3959_ | _0.7033_ | **0.8082** | 0.5928 | _0.7845_ | 0.947 | 0.6535 | _0.6979_ | 0.0642 | _11.549_ |
| [2,3,4] | 0.3908 | 0.7029 | 0.8046 | _0.5921_ | _0.7845_ | **0.949** | 0.6535 | 0.6968 | **0.0664** | **11.502** |
| 4→2 | **0.4027** | **0.7037** | _0.8058_ | 0.5908 | **0.7872** | 0.943 | **0.6614** | **0.6992** | _0.0661_ | 11.605 |
| | | | | | $k_{\mathrm{ev}}=2$ | | | | | |
| Oracle | 0.3831 | **0.6848** | **0.7758** | 0.5802 | 0.7726 | _0.945_ | 0.6330 | 0.6821 | 0.0625 | 12.225 |
| [2,3,4]→2 | _0.3865_ | 0.6827 | 0.7728 | _0.5821_ | 0.7726 | 0.944 | **0.6511** | _0.6845_ | _0.0662_ | _12.070_ |
| [2,3,4] | **0.3968** | _0.6835_ | _0.7729_ | 0.5813 | **0.7780** | 0.943 | _0.6425_ | **0.6854** | **0.0675** | **12.033** |
| 4→2 | 0.3959 | 0.6789 | 0.7716 | **0.5797** | _0.7742_ | **0.946** | 0.6385 | 0.6835 | 0.0643 | 12.132 |

