# OpenReview forum: "MoE-PHDS: One MoE checkpoint for flexible runtime sparsity"
_ICLR.cc/2026/Conference — Submitted to ICLR 2026_

### Official Review · Reviewer_ygxV · 2025-10-16

**Soundness:** 3
**Presentation:** 3
**Contribution:** 2
**Rating:** 4
**Confidence:** 4

**Summary:**

This paper proposes a technique to be able to change the sparsity level of MoE levels during inference. The technique involves an SFT phase on top of pre-trained MoE models where the top-k level (amount of sparisty) is varied initially and then is (optionally) annealed to a single top-k (k_{ev}) (where k_{ev} is usually < k_{pretrain}). Initially, they show that publicly available MoEs as well as their internal models show some degradation when the sparsity level is varied during inference - something which is expected as this introduces a train/test discrepancy. They show that with their technqiue of varied top-k followed by a k-annealing, which they call MoE-PHDS, mitigates this degradation. This technique is introduced within a follow-up SFT phase on the MoE model. The authors present results on several MoE models like OlMoE and QWEN and internal models. They compare MoE-PHDS to a baseline where they simply SFT on a single fixed K (Naive baseline) and a setting where they SFT with the same k used in pretraining (Oracle).

**Strengths:**

The main idea of the paper was clearly explained and motivated by observed performance degradation in open-source as well as internal MoE based LLMS. The paper introduces a dataset agnostic post-pretraining technique to be able to control the sparsity level of MoEs. Also, given the models and datasets they chose, the experimental settings covered the ground I'd want to see. I also appreciate the authors making sure to explain that the benefits of their technique are more pronounced on internal models as compared to publicly available models.

**Weaknesses:**

I believe their are several weaknesses in this paper.
Compared to the other baselines, the impact of MoE-PhD is quite marginal in most cases. For several models and experimental settings, the Naiva and Oracle baseline actually outperform MoE-PHDs and even when MoE-PHDs does outperform, the delta is quite small which makes me wonder how useful the technique really is.

It seems gains are more pronounced on non-optimal internal models and not on already well-tuned models which suggests this technique is less useful for well-trained models.

The Naive baseline is trained on a single k but it seems that to compare it in a fair manner to the Oracle or PHD technique, you'd use the highest K which PHD was trained on otherwise the finetuning with Navie sees less flops and is undertrained.

Considering the pre-trained models the authors evaluated on are fairly large sized, I would have liked to see benchmark performance of MOE-PHDS and baselines on more challenging tasks like MMLU. Did the authors evaluate on those datasets?

Also, I don't understand the reasoning behind the choice of SFT dataset. Why was a QA style SFT dataset chosen? Why not a large-scale pretraining corpus? It is well understood that SFT datasets can have a meaningful imapct on the models behavior and thus I'd think that to prevent that, the authors would simply SFT on a pre-training dataset and evaluate on the same standard benchmarks.

**Questions:**

"at inference time we intentionally restrict the model space (fewer experts than
used in pretraining), by declaring a smaller k after training."

What happens if we want lower sparsity during inference compared to training i.e higher top-k? The motivation being that we may want more test-time compute for harder problems. I'm curious if the authors tried MoE-PHDS or any other baseline with a k > k_{train}


Oracle finetuning regime
What's is the motivation behind fine tuning an existing model on the same K_pre? Why not just run evaluation the pre-trained model? Is it to normalize for the impact of the SFT-dataset? I suspect that SFT on top of a model trained with some alignment type techniques may worsen performance and what we may be seeing is noise from there.

Labels like Oracle 2 are confusing. Please include Oracle k-2 or some indication showing this is the sparsity level

It seems to be that since models spend 99%+ time during inference, why not just pretrain with a much higher k to begin with and introduce sparsity during inference rather than using post-training techniques. I recall [1] does that. This would give the highest level of choice over K without introducing catastrophic performance degradation (i.e when K > K_{pretrain}).

[1] Dense Training, Sparse Inference: Rethinking Training of Mixture-of-Experts Language Models

---

### Official Review · Reviewer_R6mu · 2025-10-31

**Soundness:** 1
**Presentation:** 2
**Contribution:** 2
**Rating:** 2
**Confidence:** 4

**Summary:**

This paper proposes MoE-PHDS (Post Hoc Declared Sparsity), a training method that enables runtime control of the MoE sparsity level (top-k) without retraining or architectural changes. It combines (1) multi-k training, where k is randomly sampled each step, and (2) curriculum anchoring, which stabilizes training by briefly fine-tuning at a fixed high k after initial steps. PHDS enables flexible and efficient MoE inference across diverse runtime budgets.

**Strengths:**

Practical use of MoE properties: The idea of adjusting the global sparsity level (top-k) to match limited resource or latency budgets effectively leverages the inherent strengths of MoE models and enhances serving flexibility.

Comprehensive empirical validation: The paper provides extensive experiments across multiple models and checkpoints (e.g., OLMoE-1B-7B-0125, Qwen1.5-MoE-A2.7B), demonstrating predictable accuracy-latency trade-offs and performance comparable to or exceeding oracle (fixed-k) models.

High transferability: Since PHDS is a lightweight SFT procedure requiring no architectural changes, it can be readily extended beyond LLMs to other MoE-based architectures such as VLMs.

Robustness demonstration: The work shows that pretrained MoEs are more robust to runtime sparsity shifts than commonly assumed, and quantitatively improves cross-sparsity agreement, confirming stable performance across varying sparsity levels.

**Weaknesses:**

Limited improvement over baselines: The performance on open models is relatively weak, and the paper lacks sufficient justification for why MoE-PHDS consistently outperforms oracle (fixed-k) models.

Insufficient evaluation scope: The evaluation benchmarks are limited; it would be valuable to assess whether PHDS remains effective on reasoning tasks such as math or code generation.

Marginal performance differences: The reported gains are often trivial or within noise range, particularly in Tables 4 and 5, where differences mostly appear at the third decimal place.

Architectural variance not addressed: The paper does not adequately discuss how structural differences between MoE architectures (e.g., shared vs. unshared experts in OLMoE and Qwen) affect PHDS behavior.

Missing dense-model comparison: A comparison with dense baselines (e.g., OLMoE vs. OLMo when k=1, given similar activated parameters) would provide a clearer understanding of the method’s efficiency.

Reproducibility concerns: The lack of released code and internal baseline models limits the reproducibility and independent verification of the presented results.

**Questions:**

Please see the weaknesses for the questions and suggestions.

Typo - Line 313: “() robustness to sparsity” may be “(ii) robustness to sparsity”.

---

### Official Review · Reviewer_uutY · 2025-10-31

**Soundness:** 3
**Presentation:** 3
**Contribution:** 2
**Rating:** 2
**Confidence:** 4

**Summary:**

The authors train moes with variable top-K with a curriculum SFT method and obtain a singular checkpoint (after picking the one amongst few) which can subsequently be used with varying K at inference time that the operator can set . Authors claim this method is superior to token-level adaptive routing for achieving "predictable latency" and SLA guarantees.

**Strengths:**

Strengths:
1. This produces a supplemental approach to token level adaptive routing sparsity or moe pruning.
2. The curriculum annealing is an interesting methodological contribution
3. The method does indeed cap the SLA when set, thus getting rid of latency spikes.
4. I like that the paper has done many experiments and ablations with different models.

**Weaknesses:**

Weaknesses:
1. The paper lacks comparison with any existing baselines with relevant works in adaptive sparsity or even pruned MoEs that could operate at the same budget. The main claim of the paper relies on the fact that adaptive token level sparsity may lead to subpar performance w.r.t. latency slas which is never substantiated in the paper through ablation studies or performance comparisons on existing benchmarks. Neither are performance comparisons on the existing benchmarks. Further in low K regime, in MoEs it is known to be communication bottlenecked, it is unclear what kind of lift this method will provide.

2. The authors claim that the adaptive routing focuses on token level choices hence it adds variance because the latency depends on inputs but the author shows no ablation against that method. We cannot determine whether this latency variance would be amortized over time in practice for example with batching or prompt caching or some other clever routing techniques etc.

3. From the performances in the tables we can observe that there is no significant difference between the performance of the cases with or without curriculum. As such it is difficult to understand the merit of the curriculum based training in isolation. Further the curriculum learning is just sampling k randomly during training without any deeper insights, so it's unclear what would happen indeed when the experts are highly specialized. Further No theoretical analysis or technical intuition to why the annealing would work.

4. There is no discussion of how this approach would work in the shared experts paradigm. I know it depends on the model they are testing on, maybe some thoughts here would be nice. Is it trivially extendable? what would you anneal to? This remains unclear

5. The paper does note at the limitations section that scalability to larger models is unknown. The larger models have more emergent capabilities, for example lets take a reasoning moe, an oss one lets say a qwen or even a gptoss model, it is highly unknown what happens to their reasoning ability when a further SFT disrupts their token routing. What happens to their function calling or coding abilities etc? Without any intuition it is difficult to generalize the approach.

6. In some of the comparisons with cross-k agreement, it feels a bit misleading to compare models that are trained completely separately with separate sft with different K_evs with the one that has been trained on a singular SFT run with variable sparsity training. Naturally a model trained on variable k will give more consistent answers across k than two separately trained models.

7. Checkpoint selection is a bit unclear. The authors select a checkpoint after annealing at different values of K_ev based on validation performance. Then claiming the same checkpoint works well at other k values is a bit of circular reasoning. Wouldn’t you want to make sure to select the checkpoint which on average has the best score across all required K_evs irrespective of which K_ev they are annealed to for selection?

8. The paper provides no ablations or studies examining whether the base routing distribution of the MoE is disrupted by the additional SFT on a broader range of tasks.

Minor but did not effect the score:

1.Section 4: Paragraph on Sparse MoEs seems to have abruptly ended.

2.Figure 4 is never described anywhere, perhaps this is what you describe as figure 3.5

3. Notations like PHDS [2,3,4]→3 are never formally defined which makes it slightly difficult to follow.

**Questions:**

See weaknesses. Some strong baselines and methodological clarity is indeed very crucial to help authors make their point across.

---

### Meta-Review · Area_Chair_CsBv · 2025-12-07

**Summary:**

Two reviewers recommend rejection, and another one rated it below acceptance, highlighting concerns regarding the paper’s technical novelty, experiments, and empirical validation. Additional issues include missing comparisons to relevant baselines and limited improvements
The authors did not submit a rebuttal. As a result, the reviewers’ concerns remain unaddressed. Therefore, I recommend rejection.

**Reviewer Concerns:**

The rebuttal was not submitted

**Reviewer Scores:**

The rebuttal was not submitted and therefore did not affect the reviewers initial decisions.

---

### Decision · Program_Chairs · 2026-01-26

Reject